# Situating zoonotic diseases in peacebuilding and development theories: Prioritizing zoonoses in Jordan

Jaclyn McAlester ☉¤*☯, Yuichiro Kanazawa ☉☯

Public Policy and Social Research, International Christian University, Mitaka, Tokyo, Japan

☯ These authors contributed equally to this work.
¤ Current address: Cardiology Department, Presbyterian Hospital, Albuquerque, New Mexico, United States of America
* jmmcalest@gmail.com

**Data Availability Statement:** All relevant data are within the manuscript and its Supporting Information files.

**Funding:** JM: Completed this research for her Master's thesis under funding from Rotary

## Abstract

Zoonotic diseases are projected to be a serious public threat in the coming decades. In 2016, the World Health Organization (WHO) recommended that Jordan prioritize their list of zoonoses, partially in response to the influx of Syrian refugees. We write this paper to expand the One Health framework by situating zoonotic diseases in peacebuilding and development theories in order to prioritize zoonotic diseases in Jordan. We employ an explanatory sequential mixed methods approach to create a modified version of the Center for Disease Control's (CDC) One Health Zoonotic Disease Prioritization (OHZDP) tool. We use an integrative literature review to develop a list of zoonoses to be prioritized. We expand the One Health framework by arguing health inequity is a form of violence, and thus promotion of health equity is a form of peacebuilding. We undertake thematic and statistical analyses to assess the 12 previously published OHZDP tools to evaluate necessity for change to the process given COVID-19 and the refugee situation in Jordan. In these analyses we use drivers of health indicators as measurements for peacebuilding and development, given these drivers are related to health inequities, to guide weighting of the criteria in our tool for Jordan. We apply our modified OHZDP tool to prioritize our disease list. We find it necessary to give socioeconomic factors greater consideration and to distribute weighting more evenly among all criteria within the tool when prioritizing zoonotic diseases in better reflect the Jordanian context and incorporate the refugee population. We find the priority zoonoses within Jordan to be bovine tuberculosis, brucellosis, and COVID-19, with most having a disproportionately negative impact on refugees. In Jordan's case, zoonotic diseases represent an area where promoting social equity for individuals is essential to the larger society. In this sense managing zoonoses is an area uniquely suited for peacebuilding.

## Introduction

Zoonotic diseases are diseases transmitted between animals and humans. Once spillover takes place from the reservoir animal host to a recipient human host, the pathogen can evolve and

International, as a Rotary Peace Fellow (no grant number allocated). https://www.rotary.org/en/our-programs/peace-fellowships The funders had no role in study design, data collection and analysis, decision to publish, or preparation of the manuscript. YK: The second author is grateful for financial support from the Japan Society for the Promotion of Science under the Grant-in-Aid for Scientific Research (C) 20K01595. https://www.jsps.go.jp/english/e-grants/ The funders had no role in study design, data collection and analysis, decision to publish, or preparation of the manuscript.

**Competing interests:** The authors have declared that no competing interests exist.

adapt itself to permit human to human transmission, as seen with cases of H1N1, Ebola virus, SARS, and MERS [1]. Zoonotic diseases are projected to be a continued serious public health threat in the coming decades, as it is estimated that 75% of new or emerging infectious diseases are zoonotic in origin [2].

Historically, zoonoses have been associated with poverty, given spillover often takes place at the human-animal interface and livestock contributes to the livelihoods of 70% of the world's rural poor [1]. Zoonoses tend to have low mortality rates but high morbidity; it is estimated that 20% of all human illnesses and death in the least developed countries are attributed to zoonoses [3]. Underdiagnosis and neglect of zoonotic diseases is often because they are difficult to diagnose, unevenly spread geographically, under-reported due to poor knowledge of disease symptoms in both patients and practitioners, and efforts towards zoonoses are underfunded [4].

In 2016, the World Health Organization (WHO) recommended that Jordan prioritize and create a national control plan for all zoonoses. This was mainly in response to concerns about the influx of Syrian refugees, as the WHO states, "Concern over zoonoses is high in Jordan as the increasing density of human populations resulting from the ongoing regional crisis may lead to hotspot areas and increased interaction with ecosystems that are untouched in the country [5, (p.16)]." Jordan has played a pivotal role in accepting refugees for decades, hosting refugees from Palestine since the creation of Israel in 1948, Iraqi refugees from the Gulf War in the 1990s, and most recently, refugees from the Syrian civil war [6, 7]. In fact, Jordan has the second-highest population of refugees globally compared to its population, at a ratio of 89:1,000 [8].

Prior to the Syrian refugee crisis, Jordan was already facing water scarcity, climbing unemployment, and development deficits in healthcare and education [7]. Given Jordan's small size and lack of natural and economic resources, the continued influx of refugees has placed a burden on national and local infrastructures [9]. Therefore, situating zoonoses in Jordan within its context is particularly important to direct limited resources to diseases with the potential for the greatest harm.

According to the Centers for Disease Control (CDC), "One Health is an approach that recognizes that the health of people is closely connected to the health of animals and our shared environment [10]." In 2014, the CDC created the One Health Zoonotic Disease Prioritization (OHZDP) tool. In contrast to existing prioritization processes, the OHZDP tool—a collaborative, multisectoral, and transdisciplinary approach—was developed specifically to meet the needs of those working in areas where quantitative data on zoonoses are scarce and ties between the human and animal health sectors may be underutilized. Prioritization processes using the CDC's OHZDP tool were successfully completed in primarily African countries, except for Pakistan and the United States [11].

To apply a One Health framework in the Jordanian, we are obligated to regard refugees as important stakeholders. To do so, the existing One Health frame of reference must be expanded.

Woehrle argues that health and peace are connected through social structures, such as violent conflict and social/economic inequalities and oppression [12]. Galtung defines violence as anything related to the use of unequal structures of power, such as racism, poverty, and restricted access to healthcare, which limit a person's potential. Therefore, peacebuilding must address these social inequities [13]. Health inequity is when a person does not have the opportunity to attain their full health potential due to disadvantages from social position or other socially determined circumstances [14]. For example, impoverished populations, such as refugees and livestock keepers, may be exposed to infected animals and contaminated products, may not have the ability to implement hygiene and sanitation recommendations, may not

have access to clean water, and often have limited or no healthcare access [1]. Therefore, a framework for peacebuilding must be included to implement a One Health approach in the Jordanian context.

Sen argues that development is the process of expanding freedoms for people, suggesting that once individuals are given freedom and adequate social opportunities, they can meet their own potential and better help one another [15]. This echoes Galtung's idea that violence is a limiting of one's potential, and as such, development as a means to equip people to meet their own potential can be seen as a form of peacebuilding [13, 15]. In fact, advances in development have directly addressed the impacts of poverty on health by raising living standards and improving essential services, such as nutrition and food security, access to clean water and sanitation, maternal and child health interventions, and increased access to education [3]. Thus, a framework for development must also be included to implement a One Health approach in the context of Jordan.

However, development has a complex relationship when it comes to zoonotic diseases. Humans have changed and continue to alter the environment in which animals and pathogens operate, creating new conditions for increased risk of zoonotic spillover. In many ways, humans have done so through advances in development, including changes in land use, animal production systems and domestication of animals, antimicrobial use, global trade, hunting wildlife and wildlife trade, urbanization, and the increasing global population [16]. Forced migration and movement of livestock is a recognized contributor to the emergence of infectious diseases [16] and thus carries implications for countries receiving high numbers of refugees.

We write this paper to expand the One Health framework by situating zoonotic diseases in peacebuilding and development theories in order to prioritize zoonotic diseases in Jordan. The remainder of our paper is organized as follows: we outline our methodology, including our framework for situating zoonoses into health and peacebuilding theories, describe our results, discuss our findings, connect to recent literature, and then conclude and state our study limitations.

## Methodology

We used an explanatory sequential mixed methods approach applied to a single case study of Jordan, conducted in three phases.

### Integrating zoonoses into health, peacebuilding, and development theories: A framework

Fig 1 displays how we integrate zoonoses into peacebuilding and development theories, which is the expanded framework necessary to prioritize zoonoses in Jordan, given the need to consider refugees as stakeholders. This is the lens through which we conduct our study.

Neglecting zoonoses often means neglecting diseases that impact the most vulnerable populations, populations already experiencing social inequities [1, 4, 5]. Thus, zoonoses are an area where health inequity is most pronounced. If violence is defined as limiting one's potential through unequal structures of power, and peacebuilding is the presence of social justice [13], then incorporating peacebuilding and development theories when managing zoonotic diseases is inevitable.

Development is a method of peacebuilding (Fig 1). In order to expand the One Health framework, we use drivers of health indicators as candidate explanatory variables in our statistical analyses to measure peacebuilding and development elements within the OHZDP tool and Jordan.

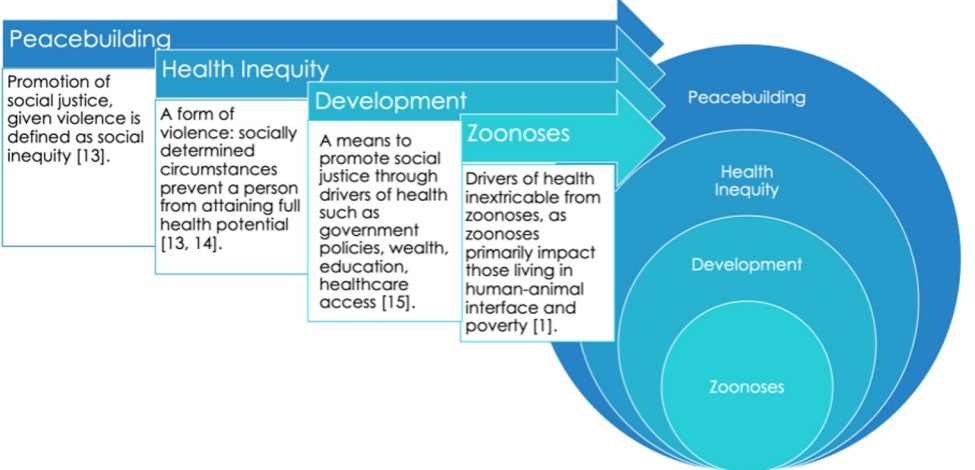

**Fig 1. Situating zoonoses within peacebuilding and health theory frameworks.**

Drivers of health include government policies, wealth, education, racial identity, gender identity, genetics, environment, medical care, health behaviors, social circumstances, and occupation [17]. This is further explained in the Phase 2 section of the methodology.

## Phase 1: Integrative literature review and producing a list of relevant zoonoses in Jordan

We conducted an integrative literature review of factors impacting the health, peacebuilding, and the zoonotic disease landscape in Jordan to produce a list of relevant zoonoses. An integrated literature review uses past research and incorporates it into a review around a new or multidisciplinary subject [18], in this case, the connection between peacebuilding, development, and zoonoses within Jordan. In addition, we gathered information regarding Jordan's experience with COVID-19 to take into consideration for our modified tool.

We collected zoonotic disease incidence data from Jordan reported in animal and human populations across a ten-year span from 2007–2016 from the central international database for reporting zoonotic diseases within the World Organization for Animal Health (OIE) [19]. This resulted in a list of 37 zoonoses. We applied two exclusion criteria to the list to establish immediacy and prevalence: 1) the disease needed to have at least one case reported in the last five years and be present for a total of five years in the ten-year span from animals and humans, and 2) the disease must have been reported in human populations at some point during the ten years. This resulted in seven zoonoses. The full list can be seen in the S1 Appendix.

However, certain zoonotic diseases do not appear on the OIE reportable diseases list. To identify other zoonoses in Jordan, we undertook a database search including: Databases within the International Christian University Library (Pubmed, JSTOR, EBSCOHost, National Center for Biotechnology Information (NCBI), Elsevier), ResearchGate, Google Scholar, the WHO, the FAO, as well as Jordan's Ministry of Health website. We used the key search terms, "Jordan," "zoonoses," "zoonotic diseases," "neglected tropical diseases," "prioritization," "prioritizing," "One Health Zoonotic Disease Prioritization tool," "refugee," "refugees," "Syrian refugees," and "Middle East."

## Phase 2: Analyses of previously published OHZDP tools

For all our statistical analyses we used R version 3.6.2 and R Studio version 1.2.5033. We conducted analyses of the previously published OHZDP tools, constructed via the CDC proposed

workshop method, to understand how zoonoses have been prioritized in the past by other, mainly African, countries. Such analyses establish a foundation for us to evaluate the necessity of changes to the OHZDP tool which better reflect the Jordanian context.

We selected the OHZDP tool due to its flexible design, which allows for the reordering of prioritization criteria within the tool so that it more accurately reflects the local context and because it is unique in that it was created with a One Health approach as its foundation [10]. One Health encourages a transdisciplinary approach [10], which is crucial in the Jordanian context where the idea of peacebuilding and development must be incorporated when prioritizing zoonotic diseases.

There are 24 past OHZDP workshops, 12 of which had reports publicly available at the time of our research. Therefore, our sample consists of 12 reports from the following countries/ regions: Burkina Faso, Cameroon, Côte d'Ivoire, the Economic Community of West African States (ECOWAS), Ethiopia, Kenya, Mali, Mozambique, Pakistan, Tanzania, Uganda, and the United States. Each country constructed its own OHZDP tool, consisting of its own five prioritization criteria, each with a single categorical question and associated weight. We pursued thematic analysis to generate patterns of meaning among those criteria and categorical questions used to prioritize zoonotic diseases among the 12 countries.

We then proceeded with statistical analyses of the criteria weighting mechanisms seen among the reports. We began with descriptive statistics of the criteria weights, which allowed us to group countries based on similarity in selection of criteria and measurement questions for further analyses of the criteria weighting. We performed logistic regression analysis (see S2 Appendix for detailed methodology). However, we found that only the intercepts demonstrated significance, suggesting that none of the explanatory variables selected for each criterion (for example, for socioeconomic and environmental criteria we selected the following drivers of health: GDP, GINI index, and median household income) explained deviation from the intercept. Therefore, we inferred the weight determination follows a more complicated mechanism and then investigated each criterion separately by means of confirmatory factor analysis.

Our thematic and statistical analyses suggested that the United States, Ethiopia, and Pakistan operated under a different framework and were therefore excluded from our confirmatory factor analysis, leaving a sample size of nine. Measurement items consisted of 16 candidate variables. We intentionally chose variables reflective of peacebuilding and development factors related to zoonoses. We did so by selecting drivers of health indicators, given drivers of health have a direct connection to health equity, as defined in our theoretical framework. We used these 16 measurement items to construct four latent factors representing criteria identified within our thematic analysis of the OHZDP tool—severity of disease, disease burden, socioeconomic and environmental, and control measures.

Some measurement items had the potential to overlap among two or more criteria. For example, hospital beds per 1,000 could represent the severity of disease or socioeconomic and environmental criteria. Therefore, we first conducted exploratory factor analysis to identify relationships so that we can assign each item to one of the criterion.

To measure severity of disease, we examined malaria incidence per 1,000; infant mortality rate per 1,000; disability adjusted life years (DALYs) lost to communicable diseases (in which zoonotic diseases are included) per 100,000; and hospital beds per 1,000. Malaria was chosen as it is a well-known zoonotic disease and both malaria incidence and infant mortality are United Nations (UN) Sustainable Development Goal (SDG) indicators [20]. Availability of hospital beds is important when considering a country's capacity for dealing with an epidemic or pandemic, and this idea is similar to the UN SDG indicator of health worker density and distribution [20]. DALYs related to communicable diseases are one of the WHO's development indicators [21].

To measure socioeconomic and environmental, we examined the percentage of population living in a rural area; sugar consumption per capita; democratic index; and percentage of GDP comprised of agriculture. The GDP percentage comprised of agriculture and the percentage of population living in a rural area are World Bank development indicators [22]. Sugar consumption has increased in developing countries and is directly related to a country's economy, as well as influenced by other key development indicators such as urbanization and trade liberalization [23]. We chose to include the democratic index as an indicator to incorporate how larger society and government function. This index is based on five categories (electoral process and pluralism, civil liberties, the functioning of government, political participation, and political culture) and these categories measure the degree of development [24].

To measure disease burden, we examined the number of cattle per capita; bovine (defined as cattle and buffalo) meat consumption per kg/capita/year; milk consumption per kg/capita/year; and mortality attributed to unsafe sanitation per 100,000. Mortality related to unsafe sanitation is part of the World Bank development indicators as well as a UN SDG indicator [20, 22]. Cattle are well known zoonotic disease hosts and transmission to humans may occur during activities at the animal-human interface such as slaughter, eating contaminated meat, or drinking unpasteurized milk [16]. In addition, these indicators are related to livestock, agriculture, and the economy, and thus represent development factors associated explicitly with zoonoses.

To measure control measures, we examined the number (x/15) of OIE surveillance measures in place to prevent bovine tuberculosis; percent of population under five sleeping under insecticide nets; number of veterinarians and para-veterinary professionals within the country; and tuberculosis mortality per 100,000. OIE surveillance measures and availability of insecticide nets indicate government policies and the capacity for implementation of policies, a key component of development. Tuberculosis incidence is a UN SDG indicator [20]. The number of veterinarians is similar to the UN SDG indicator of healthcare worker density, but specific to zoonoses. It is also an indicator of wealth and education opportunities within a country.

We scaled our data to facilitate analysis, given our measurements have a wide numerical range. We created data frames for each of the four latent factors and then applied the scale function. We conducted preliminary data analysis to test univariate assumptions using MVN version 5.8 and we fit each model using lavaan version 0.6–7 within R. There was no missing data problem to address. We used a correlation matrix rather than a covariance matrix; given the measurement units in our data vary, we want to keep the test of the loading of the first indicator for each factor, and we are interested in testing not only the direction of the relationship but also the strength [25, 26].

We used the data collected from our confirmatory factor analysis to predict a weighting mechanism for Jordan. For this process, we incorporated data from the Hofstede Cultural Typology tool. Given construction of our OHZDP tool is partly based on inferences drawn from the 12 other countries' tools, which are primarily in Africa, we found it necessary to assess cultural compatibility between Jordan and the countries, as cultural factors may influence decision-making behaviors [27]. This online tool compares countries in regards to dimensions of power distance, individualism, masculinity, and uncertainty and avoidance, rating each on a 0–100 scale [27]. We input the countries into the tool to generate a typology comparison in these dimensions. This tool did not provide data for Cameroon, Côte D'Ivoire, Ethiopia, Mali, and Uganda.

We used Stats' version 3.6.2 in R to compute the Euclidean distance between Jordan and that of the other nine countries assessed for each of the candidate variables which were found to be statistically significant, as well as data from the Hofstede tool. We excluded malaria

incidence and use of insecticide nets as candidate variables as these variables are not applicable to Jordan. When evaluating the measurement items under disease burden, we excluded countries which used this criterion twice in their tool (Cameroon, Côte D'Ivoire, Kenya, and Tanzania) as we wanted to evaluate a framework where disease burden accounts for only one of the five criteria. We used Euclidean distance so that our calculations would be based on a measurement of similarity [28] between Jordan and the other countries. We then calculated an estimated weight for each variable using the Euclidean distance. We summed the Euclidean distances to use as the denominator for each variable. We multiplied each country's associated weight with the Euclidean distance, then divided this by the sum of the Euclidean distances. All raw data and R codes can be found in the S2 Appendix.

## Phase 3: Construction and application of our OHZDP tool

We prioritized zoonotic diseases in Jordan from an objective perspective, given we are not stakeholders ourselves. Therefore, rather than use the CDC's proposed workshop process to create an OHZDP tool, we constructed our OHZDP tool based on insights from our analyses of the past 12 OHZDP tools, our expanded One Health framework which incorporates peacebuilding and development factors, and inferences from the experience of COVID-19 found in our integrative literature review. We provide further detail regarding construction in our discussion section.

We applied our OHZDP tool to the list of relevant zoonotic diseases we identified in Jordan to produce a prioritized list. All categorical questions within the tool have either yes/no answers or ordinal multinomial answers, with weights assigned to each answer. Data to answer each question was identified through extensive literature search, including the following databases: Databases from within the International Christian University library (Pubmed, EBSCO-Host, JSTOR, National Center for Biotechnology Information (NCBI), Elsevier), WHO, OIE, FAO, Research Gate, and Google Scholar. For each criterion, the score was multiplied by its associated weight. Then we summed the scores from all five categorical questions. We normalized all the final disease sums such that the highest final score was 1. Then, we ranked according to the normalized score. Scoring sheets with corresponding sources can be found in the S3 Appendix.

## Results

### Phase 1 results

We identified 11 zoonotic diseases relevant to Jordan to undergo prioritization. Leishmaniasis, bovine tuberculosis, brucellosis, echinococcosis, anthrax, rabies, and Q fever were present in both animals and humans within the last ten years in Jordan, as found in the OIE database [19]. MERS, through literature review, was also found to have human cases reported in the last ten years and to be present in camel populations within Jordan [29]. A joint WHO and China study investigating the origin of COVID-19 states the most likely pathway for introduction into human populations is through an intermediate animal host, suggesting it too, is a zoonotic disease [30]. There have been no human cases of avian influenza in Jordan since 2006; however, there have been reports within the chicken population, and Jordan itself identified it as a priority disease, even going so far as to develop a response plan [5, 31] Crimean Congo Hemorrhagic Fever (CCHF) has been reported in animal populations within Jordan, but no human cases have yet been reported. However, the WHO has indicated it is a global priority disease, and Jordan placed it on a list of reportable diseases during a surveillance project, indicating it is considered a potential public health threat [19, 32].

## Phase 2 results

Each of the 12 OHZDP tools evaluated contained five criteria with a corresponding categorical question. Different criteria were used among the tools. Based on the thematic analysis, we found six themes among the criteria and 12 subcategories among the categorical questions (Table 1).

Disease burden and severity of disease were seen among all 12 reports. Control measures was found in 11 reports and socioeconomic and environmental in 10 of the reports. The fifth criteria were the least consistent among the reports, with four countries (ECOWAS, Ethiopia, Mozambique, and Pakistan) employing bioterrorism potential and five countries (Burkina Faso, Mali, Pakistan, Uganda, and the United States) using existing inter-collaboration. Cameroon, Ivory Coast, Ethiopia, Tanzania, and Kenya selected the criteria of disease burden twice within their tool by utilizing two questions among the three categorical indicator themes as separate criteria. Detailed information of which countries correspond to which criteria and indicator subcategories can be seen in the S2 Appendix.

Table 2 summarizes the criteria weighting used by each country. Burkina Faso, Cameroon, Côte D'Ivoire, Kenya, Mali, and Tanzania used only four criteria to form the bulk of their weighting, with the sum greater than 0.90, suggesting that the fifth criteria contributed significantly less to overall weight. However, for ECOWAS, Mozambique, and Uganda, the total weights from the four criteria are less than 0.90, suggesting that the fifth criteria contributed more significantly to the total weight. Ethiopia and Pakistan both chose not to employ socioeconomic and environmental criteria at all. The United States did not utilize control measures, and yet with only three criteria (severity of disease, disease burden, and socioeconomic and environmental), the weighting is already nearly 1 (at 0.96). Median weight range between criteria was 0.14 and mean weight range was 0.1473.

Table 3 shows the results for each one-factor model fit in our confirmatory factor analysis. Results indicate what sort of drivers of health influence weighting decisions. For example, we found that the higher incidence of malaria per 1,000, the higher a country's weighting of severity of disease.

Table 4 demonstrates the results from our weight estimation using Euclidean distances calculated for each candidate variable and the cultural typology assessment. We found severity of disease and disease burden to have the highest predicated weighting, followed by socioeconomic and environmental and control measures.

**Table 1. Thematic and descriptive analyses results of past OHZDP tools.**

| Criteria | Mean Criteria Weight/ Weight Range | Categorical Question Indicator Themes |
|---|---|---|
| Severity of Disease | 0.28 0.20–0.41 | • Case fatality rate /morbidity and mortality<br>• Incidence of disease |
| Disease Burden | 0.33 0.18–0.48 | • Epidemic potential<br>• Presence of disease<br>• Modes of transmission |
| Control Measures | 0.18 0.13–0.21 | • Vaccinations/treatment availability<br>• Existence of diagnosis/ prevention/control strategies. |
| Socioeconomic and/or Environmental | 0.17 0.11–0.21 | • Solely economic indicator, such as impact on animal productivity<br>• Broad question incorporating aspects of all three: social, economic, and environmental |
| Existing Inter-collaboration | 0.09 0.04–0.19 | • Presence of existing coordination mechanisms |
| Bioterrorism Potential | 0.14 0.11–0.17 | • Pathogen listed as an official bioterrorism agent<br>• Bioterrorism potential |

**Table 2. Individual criteria contribution to total weight.**

| Countries | Severity of Disease | Socioeconomic | Disease Burden | Control Measures | Combined Weight of Four Criterion | Criteria Weight Ranges |
|-----------|---------------------|---------------|----------------|------------------|-----------------------------------|------------------------|
| Burkina Faso | 0.35 | 0.15 | 0.33 | 0.13 | 0.96 | 0.35–0.04 |
| Cameroon | 0.20 | 0.198 | 0.403 | 0.198 | 0.996 | 0.202–0.198 |
| Côte D'Ivoire | 0.2099 | 0.19355 | 0.403 | 0.1937 | 0.9971 | 0.2099–0.1935 |
| ECOWAS | 0.36 | 0.12 | 0.18 | 0.19 | 0.85 | 0.36–0.12 |
| Ethiopia | 0.23 | - | 0.41 | 0.19 | 0.83 | 0.23–0.17 |
| Kenya | 0.23 | 0.21 | 0.39 | 0.17 | 1 | 0.23–0.17 |
| Mali | 0.35 | 0.13 | 0.26 | 0.17 | 0.91 | 0.35–0.13 |
| Mozambique | 0.286 | 0.11 | 0.33 | 0.142 | 0.868 | 0.33–0.11 |
| Pakistan | 0.406 | - | 0.23 | 0.175 | 0.811 | 0.406–0.07 |
| Tanzania | 0.21 | 0.20 | 0.398 | 0.186 | 0.994 | 0.21–0.186 |
| Uganda | 0.21 | 0.19 | 0.205 | 0.205 | 0.81 | 0.21–0.186 |
| US | 0.28 | 0.156 | 0.48 | - | 0.96 | 0.33–0.077 |

## Phase 3 results

Table 5 shows our modified OHZDP tool which we used to prioritize our list of relevant zoonoses identified for Jordan.

Table 6 displays the results of our prioritization. We found the six priority zoonoses in Jordan to be bovine tuberculosis, brucellosis, COVID-19, anthrax, MERS, and avian influenza.

## Discussion

Our aim is to prioritize zoonotic diseases in Jordan under the One Health framework. To achieve that, we find it necessary to expand the One Health framework by situating zoonotic diseases in peacebuilding and development theories. Past OHZDP tools and the recently published OHZDP tool for Jordan by Kheirallah et al. [33] were constructed prior to COVID-19. However, the experiences of COVID-19 have magnified the ways zoonoses can bring grave

**Table 3. Confirmatory factor analysis results.**

| Latent Factor | SRMR | CFI | Indicator | Factor Loading | SE | Z | P |
|---------------|------|-----|-----------|----------------|-----|-----|-----|
| Severity of Disease | 0.041 | 1.000 | Malaria incidence per 1,000 | 0.902 | 0.238 | 3.792 | 0.000 |
| | | | Infant mortality per 1,000 | 0.837 | 0.250 | 3.343 | 0.001 |
| | | | Disability-adjusted life years lost to communicable diseases | 0.786 | 0.259 | 3.033 | 0.002 |
| | | | Hospital beds per 1,000 | -0.598 | 0.288 | -2.079 | 0.038 |
| Socioeconomic and environmental | 0.027 | 1.000 | Percent of population living in a rural area | 0.728 | 0.279 | 2.610 | 0.009 |
| | | | Sugar consumption per capita | 0.812 | 0.267 | 3.041 | 0.002 |
| | | | Democratic Index | 0.787 | 0.271 | 2.908 | 0.004 |
| | | | Percentage of GDP consisting of agriculture | 0.581 | 0.300 | 1.936 | 0.053 |
| Disease burden | 0.031 | 0.949 | Cattle per capita | 0.732 | 0.264 | 2.776 | 0.005 |
| | | | Bovine meat consumption per capita | 0.934 | 0.226 | 4.133 | 0.000 |
| | | | Milk consumption per capita | 0.922 | 0.229 | 4.034 | 0.000 |
| | | | Deaths from unsafe sanitation per 100,000 | 0.591 | 0.283 | 2.087 | 0.037 |
| Control measures | 0.05 | 1.000 | Number (x/15) of OIE surveillance measures prevent bovine tuberculosis | 0.680 | 0.274 | 2.481 | 0.013 |
| | | | Percent of population under five using insecticide nets | -0.859 | 0.246 | -3.493 | 0.000 |
| | | | Tuberculosis mortality per 100,000 | 0.926 | 0.233 | 3.938 | 0.000 |
| | | | Vet and paravet professionals | 0.640 | 0.280 | -2.287 | 0.022 |

*Note*. SRMR: standardized root mean squared residual. CFI: comparative fit index. SE: standard error.

**Table 4. Estimated weights for modified tool calculated from candidate variables.**

| Criteria / Average Weight | Candidate Variable | Weight |
|---|---|---|
| Severity of Disease | Infant mortality per 1,000 | 0.283 |
| 0.284 | Disability-adjusted life years lost to communicable diseases | 0.277 |
| | Hospital beds per 1,000 | 0.304 |
| | Hofstede Cultural Typology | 0.271 |
| Socioeconomic and environmental | Percent of population living in a rural area | 0.167 |
| 0.171 | Sugar consumption per capita | 0.166 |
| | Democratic Index | 0.188 |
| | Hofstede Cultural Typology | 0.161 |
| Disease burden | Cattle per capita | 0.274 |
| | Bovine meat consumption per capita | 0.276 |
| 0.270 | Milk consumption per capita | 0.263 |
| | Deaths from unsafe sanitation per 100,000 | 0.266 |
| | Hofstede Cultural Typology | 0.358 |
| Control measures | Number (x/15) of OIE surveillance measures in place for cattle to prevent bovine tuberculosis | 0.179 |
| 0.171 | Tuberculosis mortality per 100,000 | 0.175 |
| | Vet and paravet professionals | 0.181 |
| | Hofstede Cultural Typology | 0.156 |

consequences, not just in terms of mortality and morbidity, but in all spheres of life [34]. This is not just due to the disease itself, but also due to the countermeasures needed to combat the disease in the form of emergency responses.

**Table 5. Modified zoonotic disease prioritization tool to be used for Jordan.**

| Criteria /Weight | Categorical Question | Answer/Scoring |
|---|---|---|
| Severity of disease: 0.28 | What is the mortality and morbidity in humans globally? | a. >5% human CFR: 3 |
| | | b. <5% CFR: 2 |
| | | c. Disease cases documented in humans but CFR unknown: 1 |
| | | d. No documentation found: 0 |
| Socioeconomic and environmental: 0.27 | Are there economic, social, and environmental impacts? | a. All three: 3 |
| | | b. Two of the three: 2 |
| | | c. One of the three: 1 |
| | | d. No impact: 0 |
| Disease burden: 0.17 | Has this disease been present in animals or humans in the last ten years in Jordan? | a. Both: 2 |
| | | b. Either humans or animals: 1 |
| | | c. Neither: 0 |
| Control measures: 0.17 | Does Jordan have capacity for diagnosis and surveillance measures in place for this disease? | a. Both diagnostic and surveillance capacity available: 3 |
| | | b. No diagnostic, surveillance available: 2 |
| | | c. Diagnostic available, no surveillance: 1 |
| | | d. No surveillance, no diagnostic capacity: 0 |
| Existing Inter-collaboration: 0.11 | Is there a response plan in place and is there a vaccine or treatment available in Jordan? | a. A response plan exists, and a vaccine or treatment is available: 2 |
| | | b. No response plan, but there is a treatment or vaccine available: 1 |
| | | c. No response plan and no vaccine or treatment exists: 0 |

**Table 6. Prioritized list of zoonoses within Jordan.**

|  | Normalized Score 0–1 | Raw score | Infection Type | Common Reservoir Hosts |
|---|---|---|---|---|
| **Bovine tuberculosis** | 1.0000000 | 2.33 | Bacterial | Cattle, elk, bison, deer |
| **Brucellosis** | 0.9565217 | 2.28 | Bacterial | Cattle, sheep, goats |
| **COVID-19** | 0.9478261 | 2.27 | Viral | Unknown |
| **Anthrax** | 0.7739130 | 2.07 | Bacterial | Cattle, sheep, goats |
| **MERS** | 0.7739130 | 2.07 | Viral | Camels |
| **Avian Influenza** | 0.7739130 | 2.07 | Viral | Chickens |
| **Rabies** | 0.5565217 | 1.82 | Viral | Dogs, jackals |
| **Q fever** | 0.5304348 | 1.79 | Viral | Cattle, sheep, goats |
| **Echinococcosis** | 0.5217391 | 1.78 | Parasitic | Dogs, cats |
| **Leishmaniasis** | 0.4608696 | 1.71 | Parasitic | Sandflies |
| **CCHF** | 0.0000000 | 1.18 | Viral | Cattle, sheep, goats |

In Jordan, health has been a unique policy area when it comes to COVID-19. While not eligible for other public relief aid, Jordan has included the refugee population in every step of its health response, even though it has no legal obligation to do so [7]. In fact, Jordan included them in their National Health Response, allowing refugees to access national health services at the same rates as uninsured Jordan nationals [35]. It is one of the first countries to include refugees and asylum seekers COVID-19 vaccinations, offering the vaccine for free [36]. Jordan instituted similar policies in the past by offering vaccinations to refugees to maintain containment of polio, measles, and diphtheria, even though to do so was costly [7].

However, despite these efforts, COVID-19 has negatively impacted all people in Jordan, with disproportionately adverse effects on the Syrian refugee community. There has been less access to key hygiene items for refugees due to difficulty in procurement from interruptions in the supply chain and the curfew [37]. Of Syrian workers, 35% lost their jobs permanently, compared to 17% of Jordanians [38]. Public sector employees were given full pay, and the government urged private sectors to provide full salary to those working from home and at least 50% of salary to those not working due to various constraints [39]. However, refugees are in a vulnerable economic position as they are ineligible for these measures and most other Jordanian government aid [37]. Education has been an area of inequity between Jordanians and Syrian refugees during the pandemic, as 70% of Jordanian children have access to online learning, but most refugee children do not have sufficient internet access [37].

## Construction of our OHZDP tool

Our analyses of the 12 past OHZDP tools suggests that among the countries there was consensus that four criteria are necessary to use when prioritizing zoonotic diseases: severity of disease, disease burden, socioeconomic and environmental, and control measures. Therefore, we included these four criteria in our OHZDP tool. We found the basic framework of the OHZDP and past tools to be sound, and as such, we did not find it necessary to create a new criterion specifically for peacebuilding and development. However, we found a need to adjust the criteria weighting to better reflect the reality of the Jordanian context, in which consideration of peacebuilding and development is crucial.

We ranked severity of disease as the most important criteria at a weight of 0.28. This is consistent with past tools, in which either severity of disease or disease burden ranked the highest. The government of Jordan prioritized infection control over the economy during its COVID-19 response [7, 35] suggesting its policy priority that preservation of human life is paramount.

This and the mortality rates seen from COVID-19 suggest that for Jordan this should be the top ranked criteria and thus given the highest weight allotment.

We ranked socioeconomic and environmental as the second most important criteria at a weight of 0.27 and disease burden as our third criteria at a weight of 0.17. In past tools, socioeconomic and environmental was either not included or ranked as third or fourth. However, our analyses of Jordan and the on-going impact of COVID-19 suggests zoonotic diseases can have devastating social and economic impacts, and Jordan's policymakers have stated the economy is a policy priority [40]. In addition, the measurement items we found to be associated with disease burden, such as deaths due to unsafe sanitation, are less prevalent in Jordan than in the other countries assessed [41]. Therefore, we gave socioeconomic and environmental the higher weighting over disease burden.

For our fourth criteria we chose control measures at a weight of 0.17. Control measures are critical to the management and control of zoonoses. In Jordan there is a need for more aggressive control measures and surveillance [5]. Jordan demonstrates willingness and capacity for implementing more control measures in its rapid COVID-19 response [7]. We selected our fifth criteria as existing inter-collaboration at a weight of 0.11, as this is a necessary element within a One Health framework, as well as within peacebuilding and development.

Rather than assigning one or two criteria a disproportionately higher weighting as seen in six of the past tools (Table 2), we found predicted weights for Jordan should be more evenly distributed across all criteria through our weight estimation. Therefore, the range among our criteria weights is 0.28–0.11; different from many past tools where we found ranges among criteria weighting to be as high as 0.35–0.04 (Table 2). Distributing criteria weight more evenly across all criteria for Jordan better reflects how interconnected the impact of zoonoses can be across all spheres, as evidenced by the experiences of COVID-19.

## Comparison with recent research

A study published May 1, 2021, by Kheirallah et al. used the traditional workshop process proposed by the CDC to construct a OHZDP tool and prioritize zoonotic diseases in Jordan, addressing some of the concerns raised by the WHO [33]. We became aware of this study shortly after May 13, 2021, when the first author officially submitted her master's thesis, based on which we write this paper. Table 7 outlines how our results for Jordan differ from those of Kheirallah et al., given we used a different framework and construction process.

Kheirallah et al.'s research exemplifies zoonotic disease prioritization conducted by Jordanian experts prior to COVID-19 as shown in Table 7 [33]. Our findings add to Kheirallah et al.'s technical guidance to a One Health approach with several notable differences. For example, Kheirallah et al. weighted socioeconomic as the lowest criteria at 0.08 [33]. However, we find our broader framework requires this criterion to be considered more prominently. In addition, our prioritized lists revealed the top three priority zoonoses in Jordan to be bovine tuberculosis, brucellosis, and COVID-19, whereas Kheirallah et al.'s top three were rabies, MERS, and avian influenza [33].

## Peacebuilding and zoonoses in Jordan

Jordan is an upper-middle-income country with an arid climate, where agriculture accounts for only 4.9% of the GDP [42, 43]. In Jordan, disease surveillance and zoonotic disease reporting are limited, and surveillance in the animal sector is low [44]. Syrian refugees have a higher communicable disease burden than Jordanian citizens, for example, tuberculosis, leishmaniasis, and brucellosis. Given the long incubation period of these infections, refugees have a high risk of carrying them into neighboring countries [45].

**Table 7. Comparison of our research and Kheirallah et al.**

|  | Our Study | Overlap | Kheirallah et al. [33] |
|---|---|---|---|
| **Aim** | To prioritize zoonoses in Jordan by situating zoonotic diseases within peacebuilding and development frameworks. | Prioritizing zoonoses in Jordan | Prioritizing zoonotic diseases of national significance to Jordan and identifying future recommendations and action plans. |
| **Theoretical Framework** | In addition to One Health and health perspective, we include Galtung's peacebuilding theories, Sen's theory of development, and the concept of health inequity with focus on impact of and on refugees | One Health | One Health; with greater emphasis on health perspective |
| **Construction of Tool** | Used thematic and statistical analyses of past tools, experiences of COVID-19, and measurement items reflecting peacebuilding and development to construct a modified OHZDP tool | Used OHZDP framework | Created OHZDP tool through CDC proposed workshop process |
| **Tool Criteria & Weight** | Severity of disease: 0.28 | 4/5 similar criteria | Severity of disease: 0.40 |
|  | Socioeconomic and environmental: 0.27 |  | Epidemiological profile: 0.22 |
|  | Disease burden: 0.17 |  | Potential transmission: 0.17 |
|  | Control measures: 0.17 |  | Availability of intervention: 0.13 |
|  | Existing inter-collaboration: 0.11 |  | Socioeconomic: 0.08 |
| **Prioritized List** | Top three: bovine tuberculosis, brucellosis, and COVID-19 | Top five include: avian influenza, rabies, and MERS | Top three: rabies, MERS, influenza |
|  | Did not include food-borne illnesses |  | Incorporated food-borne illnesses and regional perspective with diseases such as malaria, dengue fever, and West Nile virus. |

Brucellosis was reported steadily throughout the ten years in our research, with the last number of cases at 414 in 2016 [19]. Conflict within the Middle East has resulted in large numbers of humans, animals, and animal products moving across borders while at the same time causing a disruption in control measures for brucellosis, leading to a re-emergence of cases in Jordan. In fact, there has been a five-fold increase in the number of human brucellosis cases from 2012–2016 [45]. Notably, Petersen et al. found that Syrian refugees carried higher rates of brucellosis than their citizen counterparts, indicating vulnerable populations within Jordan may be more at risk [45]. Though, the actual burden of brucellosis in Jordan is likely unknown, as it is underdiagnosed and underreported among pastoralists due to barriers of inaccurate diagnosis and lack of surveillance [46]. The case of brucellosis in Jordan leads us to infer Syrian refugees may also have higher incidences of other zoonoses on our list, though evidence is circumstantial.

According to the WHO 2019 Global Tuberculosis Report [47] it is estimated that there were around 143,000 cases of bovine tuberculosis in humans in 2018, with about 12,500 deaths. In Jordan, the most recent numbers document 614 cases in 2014 [19]. The WHO states that mitigating the risk of tuberculosis to humans cannot be achieved without controlling bovine tuberculosis in humans and animals and improving food safety [47]. Jordan had set a goal to eradicate tuberculosis by 2025; however, they have since postponed this due to the impact of the Syrian refugees on case numbers and the healthcare system. There is a lack of data regarding which populations are more impacted by bovine tuberculosis within Jordan, though given its association with consuming unpasteurized animal products, livestock keepers, and poverty [48], it would not be surprising if refugees carry a higher burden of this disease. This suggests there is a social justice component in addressing bovine tuberculosis.

Rabies is a viral zoonotic disease primarily transmitted to humans through dog bites or scratches. It is present globally, though disproportionately impacts low-income and rural areas, which account for 80% of human cases. Given this, we infer refugees may be at higher risk. Unfortunately, once symptoms appear, rabies is nearly 100% fatal [49]. The main sources of rabies in Jordan are stray dogs and jackals. Jordan and Israel launched a joint effort to

manage rabies in strays by using baits with the oral vaccination, distributing them by airplane at the border [50]. This demonstrates how implementing control measures for zoonotic diseases can be an area for international cooperation.

There are about 15–18 thousand camels in Jordan [51]. Camels are known carriers of MERS. MERS belongs to the family of viruses known as *Coronaviridae*, the same as COVID-19. Unlike many other zoonotic diseases, MERS quickly developed the capability for human-to-human transmission, and while cases from 2012–2019 have remained low at 2,442 people, the mortality rate is devastatingly high at 35% [52]. Doremalen et al. conducted a study of two geographically separated camel herds in Jordan. They confirmed circulation of MERS within the camel populations, suggesting that though earlier outbreaks of MERS within Jordan were linked to travel-related activities, there is a real possibility of spillover into the human population at the local level from camels [29].

The fact that we found brucellosis and bovine tuberculosis to be the top two priority zoonoses list is a result of our expanded One Health peacebuilding and development framework, which allows for a more inclusive approach to the pluralistic population of Jordan. Ultimately, zoonoses are an issue in Jordan that represent an area of health inequity and there is a need for improved surveillance and management systems. In Jordan, if health inequity is a form of violence by Galtung's definition [13], and zoonoses represent an area of health inequity, then managing them is a form of peacebuilding. In Jordan's case, refugees must be considered when it comes to health and specifically zoonotic disease measures.

## Conclusion

We produced a prioritized list of zoonotic diseases for Jordan, with the top six priority diseases found to be: bovine tuberculosis, brucellosis, COVID-19, rabies, anthrax, and MERS. This contributes to a gap in the literature and the WHO's recommendation of prioritizing zoonoses within Jordan. In addition, it provides a foundation to guide the concentration of initial capacity-building efforts around a few critical diseases likely to have the greatest consequences by focusing limited resources and encouraging collaboration and planning across all relevant sectors [11]. To the best of our knowledge, we are the first study to connect peacebuilding and prioritizing zoonotic diseases, as we adjusted our weighting mechanism based on peacebuilding and development indicators, specifically, drivers of health relevant to Jordan and zoonotic diseases.

## Study limitations

The sample size of published OHZDP reports available was small at 12, making it difficult to find significant relationships within the data and limiting the generalizability of the results. In addition, past OHZDP tools are overly represented by sub-Saharan African countries, which suggests the need for zoonotic disease prioritization processes in other regions, such as Southeast Asian countries, where the threat posed by zoonoses is prevalent, and at the same time peacebuilding efforts are considerably endangered. In Jordan much of the data on refugees is overly represented by Syrian refugees or does not specify which refugees are impacted (i.e., Pakistani, Iraqi, etc. . .). Therefore, we were unable to consider the full impact of all refugee populations. In addition, data collected may be overrepresented by camp refugees, as it is difficult to discern whether data collected in the urban areas (where 80% of the refugee population lives) applied to Jordanians or refugees. Lastly, there is a gap in literature and databases regarding the incidences of zoonoses within Jordan, which suggests the need for further research.

## Supporting information

**S1 Appendix. Phase 1 raw data.**
(DOCX)

**S2 Appendix. Phase 2 raw data, datasets, and R coding.**
(ZIP)

**S3 Appendix. Phase 3 raw data.**
(DOCX)

## Author Contributions

**Conceptualization:** Jaclyn McAlester, Yuichiro Kanazawa.

**Data curation:** Jaclyn McAlester.

**Formal analysis:** Jaclyn McAlester, Yuichiro Kanazawa.

**Investigation:** Jaclyn McAlester.

**Methodology:** Jaclyn McAlester, Yuichiro Kanazawa.

**Project administration:** Jaclyn McAlester.

**Supervision:** Yuichiro Kanazawa.

**Writing – original draft:** Jaclyn McAlester.

**Writing – review & editing:** Jaclyn McAlester, Yuichiro Kanazawa.

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
