## [Decision Letter · Decision Letter 0]

6 Jan 2022

PONE-D-21-32083Addressing zoonotic diseases in Jordan: the inextricable link between health and peacebuildingPLOS ONE

Dear Dr. McAlester,

Thank you for submitting your manuscript to PLOS ONE. After careful consideration, we feel that it has merit but does not fully meet PLOS ONE’s publication criteria as it currently stands. Therefore, we invite you to submit a revised version of the manuscript that addresses the points raised during the review process.

We look forward to receiving your revised manuscript.

Kind regards,

Rebecca Lee Smith, D.V.M., M.S., Ph.D.

Academic Editor

PLOS ONE

Journal Requirements:

Reviewers' comments:

Reviewer's Responses to Questions

**Comments to the Author**

1. Is the manuscript technically sound, and do the data support the conclusions?

Reviewer #1: Partly

Reviewer #2: Yes

2. Has the statistical analysis been performed appropriately and rigorously? 

Reviewer #1: Yes

Reviewer #2: Yes

3. Have the authors made all data underlying the findings in their manuscript fully available?

Reviewer #1: Yes

Reviewer #2: Yes

4. Is the manuscript presented in an intelligible fashion and written in standard English?

Reviewer #1: No

Reviewer #2: Yes

5. Review Comments to the Author

Reviewer #1: General Comments

The authors work is an interesting that they tried to develop a customized tool consisting of various relevant criteria used for prioritizing zoonotic diseases by considering the impact of violence and migration of people on the emergence, occurrence and spread of Zoonotic diseases within the context of Jordan situation. A combination of different methods was used, and the manuscript is written well and exhaustively. The authors also provided supplementary supporting files. However, there are many points that need further clarification before publication. My assessments, questions, and suggestions are provided here as follows point by point under each section of the manuscript which might the authors can consider during the revision of the manuscript.

Title

The term “addressing” is too generic and not reflecting the content and essence of the study. Addressing zoonotic diseases technically connected with intervention which might be prevention and/or control of diseases with measurable actions by prioritizing the diseases and allocating available resources. The overall aim of the study is just prioritizing zoonotic diseases by considering the role of peace building, health and COVI-19 in the list of zoonotic diseases in addition to the criteria used by other OHDP tools. However, the role of health and peace in prioritizing zoonotic diseases within the framework of the new modified tool was not considered and the linkage is not clear. Consider modifying the title.

Abstract

The author should clearly state the objective of the study after providing 1-2 statements that justify the main reason(s) for conducting this study.

-Lines 25-29 – It seems that the main aim of the study is providing a conceptual framework that situates health and peace building for prioritizing the zoonotic disease. But, elsewhere in the manuscript, it is stated that the aim is to prioritize zoonotic diseases using modified OHZDP as a tool. Thus, the author should clearly state the objective of the study after providing 1-2 statements that justify the main reason(s) for conducting this study.

-The authors are highly skewed at the beginning that health and peace building and Covid 19 can change the proposed and commonly used OHZDP process. I think these factors (Health and peace building) should be considered during criteria setting and weighing the criteria for evaluations of the selected diseases based on the literature and expert opinion for prioritize zoonotic diseases. Covid 19 should be treated as one disease and evaluated against these criteria (health and peace) and the other criteria. If the interest is to show the role of health and peace building, the author should indicate the degree of weight allocated to them during criteria weighing and why these bear the higher/highest weight under Jordan Situation. this information is not mentioned in the manuscript.

Introduction

Generally, it is too long and is very difficult for the readers to easily understand the intent of the study. Consider the following points for revision:

-As you did in the Abstract, begin with highlighting about the zoonotic diseases and then, make linkage with peace building, health inequity and development followed by rationale for designing and conducting this study within the context of Jordan. Here, you should vividly justify or argue why developing new proposed tool is needed in comparison to the OHZDP suggested by WHO that have been used in many countries.

-Condense paragraphs 1-4 to 1-2 paragraphs by only showing the interlinkage of peace building, health inequity and development with zoonotic diseases. As long as you cited the relevant references, the theoretical background can be left for readers for further reading.

- line 60.-Is “she” refers to “ Woehrle 3”. If that is the case, 3 represents CDC website. Cross check the citations and references across the document. After correction, replace “she” with “The author”.

Lines 97-105- Combine this paragraph with the last paragraph (line 136-148) as a rationale for this study.

Lines 86-88 and 115-122- In lines 86-88, animal is mentioned as a reservoir for COVID 19 and considered as a zoonotic disease with no supporting evidences from the literatures. In contrary to this assumption, you mentioned that the role of animal as intermediary host is yet to be identified and established. Which one is true? Under these scenarios, how you can argue the limitation of a study by Kheriallah and considered COVID 19 in the list of Zoonotic diseases? Can you provide compelling evidences particularly in Jordan that supports animals are one of the sources of COVID 19 infection for humans and vice versa to consider it as a zoonotic disease? Mind that your focus is zoonotic diseases!!!

Lines 106-107- it is interesting that the authors scholarly acknowledged the work of Kheirallah et al 2021 to avoid release of overlapping publications with similar aims from one country within short period of time. You mentioned your work is conducted independently and you are interested to consider the prioritization of zoonotic diseases by including COVID 19 which was the limitation of Kheirallah et al. I do not think that this should not be a justifiably reason to initiate and conduct this study. If you think the work is independent, you should not raise this point, Kheirallah et al work, in the introduction as a justification for designing and conducting this study which is not actually the case. The point here is why you did not consider this work as one tool in the analysis of previously published OHZDP tools once you knew the publication before submission of this manuscript for publication. Rather, as you mentioned and discussed it should be addressed in the discussion section.

-Lines 123-135: The justification of considering the potential impact of increasing in number of refugees on the increasing risk of zoonotic infection is important rationale for this study which makes Jordan situation peculiar.

-Can you explain how this study overcomes the limitation on the lack of data regarding zoonotic diseases in Jordan mentioned by Kheirallah et al. In this study, you also mentioned under study limitations lack of data on the relationship between health and peace building on the incidence of zoonoses in Jordan.

-Line 127. You argued that ….the groups are not inclusive enough to comprehensively represent the spectrum of issues in Jordan as the experience of refugees was not considered? If not inclusive, which sector/organization(s) should have been considered to address this gap? How your study overcome this gap?

Line 147- delete the last sentence.

Methodology

-As mentioned in the introduction, the potential rationale for this study is the under representation of the role of health and peace which can potentially influence the rank of priority zoonotic diseases in Jordan. You justified importance of the use of the proposed OHDP in countries with limited data on zoonotic diseases. Why you did not use the ODHP by considering this criteria and providing evidences, of course as a complementary to study Kheirallah et al, that show the significance of the criteria may be by allocating high weight which can potentially shift the rank of zoonotic diseases in Jordan? I do not see the significance of the thematic and statistically analysis of the previous tool used in different countries to see the similarity and predict the situation in Jordan. One of the significance of the use OHZDP is prioritizing the zoonotic diseases within the context of each country by involving the key stakeholders from various sectors and establishing inter-sectorial cooperation to combat the priority diseases.

- It would be nice if you separately present the data analysis i.e. pull all the analyses under one sub headlining-Data analysis.

Line 165: To talk about sample size there should be a sampling frame from which the sample can be drawn based on selected probability or non-probability sampling technique. How, the 12 tools were selected? This is also mentioned in the limitation section. Were only 12 countries prioritized zoonotic diseases using OHDZP? The general framework in all of the 12 countries is similar in using the OHDZP with minor difference in the number of criteria and measurement scales. Can we say that they are different tools?

Line 169: We then,

171: The framework is the same for all where they used the WHO proposed OHZDP. Replace framework as”……groups countries based on similarity in selection of criteria and measurement questions….”

172. Put “to be operating under a similar framework” after Tanzania. Six countries,…

177: What are those three category intercepts? Specify them.

178-180: Why this is possible in practical sense?

181-182: Re-phrase the sentence and clearly mention the dependent and independent variables for the logistic regression analysis performed. Specify the three categories and the baseline category.

191-193: Re-phrase the sentence.

196: How the data was scaled? Add a brief description.

210-213: Only incidence data? No prevalence? Is it possible to indicate how the incidence of the diseases were diagnosed and reported (clinical, laboratory, confirmation?)

-citation for OIE database

214-216: What the “immediacy” really refers in terms of measurement of diseases? Time?

-1) Delete “the disease needed to have” then, add “from animals and humans” after….span,…..

2) ….reported in humans and animals population.

217: Delete “ this resulted in in seven zoonoses” from here and take to result section.

218-221: Including but not limited? this is confusing and please add the other search databases searched for transparency. Did you use any systematic way to search from these databases? Any key search terms used during search? Selection criteria? All these are worth mentioning.

223: …this? you mean data on list of zoonoses? Mention what “ this “ refers. Here you are in another heading ( Pahse3).

Lines 224-225: The statement is not complete and needs re-phrasing. Indicate for what purpose the data was used. To construct modified zoonotic diseases prioritization tool? What is the central point for evaluating for the cultural compatibility between Jordan and other countries given the objective of the study (impact of violence on health)? What is the advantage over the stakeholder opinion based prioritization? Add citation for the data collected.

Line 235: the experience of covid-19? Is it specifically mentioned in the methodology? Covid 19 should be treated as one of the diseases and evaluated based on the identified criteria and weight given.

-Start with we constructed a modified zoonotic prioritization tool

Line 240-242: Again- including but not limited? this is confusing and please add the other search databases searched for transparency. Did you use any systematic way to search from these databases? Any key search terms used during search? Selection criteria? Provide website addresses for each database searched.

Line 242- The criteria and the questions were identified based on the thematic and descriptive analyses. But, it seems that the authors answer (score) each questions of measurement of the criteria for each disease based on literature. I do not think that the authors can get clear information to score each questions. It would have been nice if this was validated by the stakeholders.

Line 246: Just before the result, mention how your question of peace/violence is addressed in the modified tool to generate a new list of priority diseases.

Line 254: Indicate that the used criteria in all countries were not the same.

Line 255: Based on the thematic analysis, we found six criteria and 12 categorical questions (Table 1).

Lines 258-259: “ This suggests……….zoonotic diseases” Take to discussion section.

Lines 259-261: Mention the countries. The fifth criteria, bioterrorism potential,….four countries (XXXX) and five countries (XXXXX).

Lines 267-273: move to discussion section.

Line 274: Rephrase: and start as “Table 2 summarizes the criteria weighting used by each country.

Lines 278-280. Avoid interpretation in the result section and consider it discussion.

Line 279: Significantly (XX? quantify).

Line 286: Revise the table by adding row/column for the weight allotted for each criterion by each respective country and also mention in the methodology how the total weight is computed.

Line 290- Delete confirmatory factor analysis and rephrase as “ Table 3 shows the confirmatory factor analysis results for each one factor model fit…….”

Line 293: Table 3: add footnotes for SRMR, CFI& SE

Line 296: No heading for phase 1 results. Be consistent.

Lines 297-366: This is the result section not discussion. Present briefly in one paragraph the identified relevant zoonotic diseases and delete the detail description from here. You can provide the citations for each disease as a supplementary file. You can take some of the information from here and use them when discussing the results of phase 3 for the top priority diseases.

Lines 360-366: These should be rather part of the methodology not the result.

Line 367: Describe the major findings summarized in Table 4 in text before the table and refer to Table 4 for detail.

Lines 370-372: Already mentioned in the methodology. Here mention about the new tool focusing the findings as a result of the modified criteria and measurement/questions…..

-Be consistent for using questions Vs categorical variables Vs candidate variables.

Lines 375-385: Take to the discussion section.

Line 389: Again the same question: Now it is clear that the criteria, the weight and the scoring of the scale for each question are based on studies in previous country according to your thematic analysis. How the authors assigned a score for each disease and prioritized the diseases in absence of stakeholder validation?

Line 391: Discussion next to Table 6

I would suggest that the authors treat the discussion section separately as a main heading by take into account the following points in sequential manner:

1.Re-iterate the purpose of the study in the first paragraph

2.Followed by discussing about the new modified tool and its advantage over the Keheiriela et al study that based on the WHO proposed tool

3.How the health and peace building issues are important in prioritizing zoonotic diseases under the Jordan situation

4.Description about the relevant zoonotic diseases in Jordan and detail discussion only for the top priority by implicating the need for targeting these diseases to initiate prevention and control.

Lines 443-454: It is not clear how the authors try to address the impact of peace/migration on the occurrence, emergence and spread of zoonotic diseases. In the modified OHZDP, there is no data summarized in the result section that shows the importance of including drivers of health linked with violence. Can you explain this a bit further?

Line 457: The issue of health is should e situated along with the other factors such as economic crisis when it comes to the case of refuges.

Line 466: Delete the discussion from here and move some of the information back to the discussion section where it fits and make your conclusion remarks separately.

Lines 472-275: I do not agree with this statement. I argue that 1) the weight allotted to each criterion cannot be considered uniform for each disease, and 2) your modified tool is also totally depended on the previously proposed tool used by different countries. May you further explain what are you communicating here?

Reviewer #2: A much appreciated interdisciplinary approach to focus on OneHealth and Refugees at one time. The way COVID has but us all at risk makes it difficult for governments in developing countries to see things under the OH theme. That being said, it is critical to focus the attention on refugees first in the introduction and to provide number of refugees in Jordan. Since 2010, Jordan has accepted around 1.5 million Syrian refugees (600,000 are officially registered with UNHCR and the rest are not registered). Three main camps are in Jordan for Syrian refugees: Zaatri, Azraq are the main two within local populations, or within a short distance. Zaatri was built and made ready in days and is not organized as a camp as that for Azraq. There are real issues with clean water, sewage, and essential services. This is a real impact on OH and peacekeeping. Especially when knowing that 80% of Syrian refugees live outside camps within local (host) community. Therefore, the results may be biased towards camp residents as data on host community refugees are extremely limited. This is another major limitations for this study.

Jordan has also accepted Iraqi refugees, and small proportions of refugees from Yemen, Libya, among others. How would this be considered into the results of the study. Also, Jordan has Palestinian refugee camps established since the 70s. Such camps have the same crowdedness and lack of clean water and sewage systems. Yet, their impact is not considered in the paper. This is another limitation for this study.

Apparently, These limitations are much needed.

Line 415: Increase in number of TB may be due to available funding to study the disease and provide surveillance activities. How would this change the study results?

line 466: a new zoonotic priority list could be easily established in Jordan using the same tool in Kheirallah's study. SO this statement is overestimation of the results. I think the sentence should be that this is the first tool to connect peacekeeping to zoonosis.

Line 472: Our list was based on experts within the OH field, especially MoH and MoA. They even changed the final prioritization by moving some diseases up the list based on their expertise in the field. This was not part of the presented results as it depended on theoretical experience not on-the-ground one.

In table 7, under Construction of Tool, what we utilized in Kheirallah et al was a standardized CDC workshop tool that prioritized Zoonotic diseases in Jordan. This has an advantage of comparing countries' priority list and allow standardized methods for comparing results over time.

Table 7: Prioritized List. We actually used regional prospective as Jordan is really connected to Syria, Iraq, Egypt and the other Arab states including Saudi Arabia, Qatar, and UAE. So our initial list was regional based on national needs.

Line 495: Socio-economic were a major concern during the workshop, Still, we did not see a major effect as this was conducted before COVID. If things were to be done today, SE would be a major issue and will have a higher weight.

Line 507: great point. Can you further elaborate and include in the major conclusion.

Another limitation is the utilization of data to withdraw data on zoonotic diseases in Jordan. KNowing that such data is not complete, not funded, does not have properly funded surveillance tools, makes it hard to rely on the available data to draw conclusions.

One major comment from working with OH in Jordan is the lack of ministry of Environment in any of the OH related activities. MoH has an environmental health unit and it covered all activities related to the environment. This is a major issue that surfaced during COVID. As such, Jordan recently established the Jordan CDC. The center is expected to take the lead in OH.

In the presented manuscript, the authors based their theory on numbers reported by multiple partners including Jordan MoH. During the workshop we conducted in Jordan, it was clear that surveillance data on zoonotic diseases are lacking in Jordan. For example, zoonotic diseases are treated in the camp and never make it to MoH but rather to UNHCR. The health system within the camp is not related to MoH directly except for COVID, which is a recent approach given the political covid issues.

In the tool that was used in Jordan, the CDC provided a guideline for which participants actually provide the 5 core questions and their scoring system. So the questions and their answers were as provided by the local stockholders. Not sure how would this change, if any, the scope of the statistical analysis used.

For Hofstede Cultural Typology, what data is provided? can you provide examples of data?

6. PLOS authors have the option to publish the peer review history of their article (what does this mean?). If published, this will include your full peer review and any attached files.

Reviewer #1: No

Reviewer #2: **Yes: **Khalid A Kheirallah

---

## [Author Response · Author response to Decision Letter 0]

13 Feb 2022

Please see the attached Response to Reviewers document. However, we have copied that document here for convenience. 

Reviewer 1

The authors work is an interesting that they tried to develop a customized tool consisting of various relevant criteria used for prioritizing zoonotic diseases by considering the impact of violence and migration of people on the emergence, occurrence and spread of Zoonotic diseases within the context of Jordan situation. A combination of different methods was used, and the manuscript is written well and exhaustively. The authors also provided supplementary supporting files. However, there are many points that need further clarification before publication. My assessments, questions, and suggestions are provided here as follows point by point under each section of the manuscript which might the authors can consider during the revision of the manuscript.

• Thank you for taking the time to provide detailed assessments, questions, and suggestions. We appreciate your generous comments. 

• We apologize for our lack of clarity; we will address all your concerns below. One point we would like to clarify first is that rather than develop a customized tool, we used a modified process to construct our own OHZDP tool. Therefore, we used the OHZDP tool framework, however, rather than constructing this through a two-day workshop with experts, we constructed a tool based on analyses of past OHZDP tools, what we’ve seen with COVID-19, and through incorporation of peacebuilding and development theories and evaluation of drivers of health as measurement items. We will explain further and address each of your comments in detail. 

Title

The term “addressing” is too generic and not reflecting the content and essence of the study. Addressing zoonotic diseases technically connected with intervention which might be prevention and/or control of diseases with measurable actions by prioritizing the diseases and allocating available resources. The overall aim of the study is just prioritizing zoonotic diseases by considering the role of peace building, health and COVI-19 in the list of zoonotic diseases in addition to the criteria used by other OHDP tools. However, the role of health and peace in prioritizing zoonotic diseases within the framework of the new modified tool was not considered and the linkage is not clear. Consider modifying the title.

• Agree. We have modified the title in order to clarify the link. The new title is: Situating zoonotic diseases in peacebuilding and development theories: prioritizing zoonotic diseases in Jordan. 

Abstract

The author should clearly state the objective of the study after providing 1-2 statements that justify the main reason(s) for conducting this study.

• Agree. We have made this revision, please see lines 24-28. 

-Lines 25-29 – It seems that the main aim of the study is providing a conceptual framework that situates health and peace building for prioritizing the zoonotic disease. But, elsewhere in the manuscript, it is stated that the aim is to prioritize zoonotic diseases using modified OHZDP as a tool. Thus, the author should clearly state the objective of the study after providing 1-2 statements that justify the main reason(s) for conducting this study.

• Agree; we have clarified our objective as such: “We write this paper to expand the One Health framework by situating zoonotic diseases in peacebuilding and development theories in order to prioritize zoonotic diseases in Jordan.” See lines 26-28.

-The authors are highly skewed at the beginning that health and peace building and Covid 19 can change the proposed and commonly used OHZDP process. I think these factors (Health and peace building) should be considered during criteria setting and weighing the criteria for evaluations of the selected diseases based on the literature and expert opinion for prioritize zoonotic diseases. Covid 19 should be treated as one disease and evaluated against these criteria (health and peace) and the other criteria. If the interest is to show the role of health and peace building, the author should indicate the degree of weight allocated to them during criteria weighing and why these bear the higher/highest weight under Jordan Situation. this information is not mentioned in the manuscript.

• We feel that the experience of COVID-19 obligates us to reconsider how diseases have been prioritized within the One Health framework in the past. In addition, we feel that the incorporation of peacebuilding and development theory is necessary for Jordan, given the unique refugee situation, but also these could be important for many other countries as well. We did not use peacebuilding and development as criteria themselves. We found the overall framework of the OHZDP tool is sound. Therefore, first we expanded the conceptual framework of One Health by situating zoonoses into using peacebuilding and development theories. This allows us to use development indicators, specifically drivers of health, as measurement items for peacebuilding. We then used these indicators to include relevant information to development and peacebuilding in guiding our weighting of each criterion. Therefore, all our weighting and criteria are influenced by peacebuilding and development indicators, rather than creating a new, singular criterion.

Introduction

Generally, it is too long and is very difficult for the readers to easily understand the intent of the study. Consider the following points for revision:

-As you did in the Abstract, begin with highlighting about the zoonotic diseases and then, make linkage with peace building, health inequity and development followed by rationale for designing and conducting this study within the context of Jordan. Here, you should vividly justify or argue why developing new proposed tool is needed in comparison to the OHZDP suggested by WHO that have been used in many countries.

• Agree. We have restructured our introduction as follows: highlighting zoonotic diseases, rationale for choosing Jordan, and then linking peacebuilding, health inequity, and development. We have chosen to discuss Jordan before peacebuilding and development as we want to emphasize why it is important to prioritize zoonotic diseases in Jordan specifically, and then discuss our arguments as to why peacebuilding and development must be considered when modifying the OHZDP tool to use in this context. We find great value in the OHZDP tool. However, given our backgrounds as a Rotary Peace Fellow (JM) and former Associate Director of the Rotary Peace Center at International Christian University (YK) we viewed the problem through a different lens. Zoonotic diseases are an issue that disproportionately impact impoverished populations and have a connection to migration; we feel an obligation to expand the OHZDP framework by incorporating peacebuilding and development elements, not just for Jordan, but hopefully for consideration for other countries with large refugee populations as well. 

-Condense paragraphs 1-4 to 1-2 paragraphs by only showing the interlinkage of peace building, health inequity and development with zoonotic diseases. As long as you cited the relevant references, the theoretical background can be left for readers for further reading.

• Agree. We have shortened some of the theoretical information provided in our original introduction. 

- line 60.-Is “she” refers to “ Woehrle 3”. If that is the case, 3 represents CDC website. Cross check the citations and references across the document. After correction, replace “she” with “The author”.

• Thank you very much for catching this error; we apologize for the inconvenience. We have restructured our introduction, so this may no longer apply. However, we have made sure to cross check our references and make the appropriate corrections. 

Lines 97-105- Combine this paragraph with the last paragraph (line 136-148) as a rationale for this study.

• Agree. We have combined these points as rationale for the study, now presented in the introduction in lines 60-74.

Lines 86-88 and 115-122- In lines 86-88, animal is mentioned as a reservoir for COVID 19 and considered as a zoonotic disease with no supporting evidences from the literatures. In contrary to this assumption, you mentioned that the role of animal as intermediary host is yet to be identified and established. Which one is true? Under these scenarios, how you can argue the limitation of a study by Kheriallah and considered COVID 19 in the list of Zoonotic diseases? Can you provide compelling evidences particularly in Jordan that supports animals are one of the sources of COVID 19 infection for humans and vice versa to consider it as a zoonotic disease? Mind that your focus is zoonotic diseases!!!

• Thank you for this excellent point. Given we have removed some of our discussion regarding Kheriallah et al.’s study, some of this may no longer apply. 

• However, we do want to provide literature to support COVID-19 as a zoonotic disease, as it is one of the diseases we evaluate. To do so, we have several sources to support this decision. The first is from the WHO convened global study of origin of SARS-CoV-2, a joint WHO-China study (see citation number 30) in which they state the most likely pathway for introduction into human populations to be through an intermediate animal host. The second source (Bashor, 2021) indicates that SARS-CoV-2 has been found in multiple animal hosts, including minks, dogs, and cats, around the world. This is being tracked by the OIE. These findings support the belief from the WHO that SARS-CoV-2 likely spilled over from an intermediate host. For Jordan, we have not found documentation of SARS-CoV-2 in an animal host within the country, however given the prevalence of MERS in camel populations (see citation number 53) and the fact that SARS-CoV-2 shares 96% homology with beta coronaviruses isolated from multiple species of bats in the genus Rhinolophus (OIE, 2020), bats which are native in Jordan (Cordova, 2007) we think the possibility cannot be ruled out. 

o For reference: 

Bashor 2021: https://www.pnas.org/content/118/44/e2105253118

OIE 2020: https://www.oie.int/fileadmin/Home/eng/Our_scientific_ expertise/docs/pdf/COV-19/1st_call_COVID19_21Feb.pdf 

Cordova 2007: Cordova, C. (2007). Millennial landscape change in Jordan. The University of Arizona Press. Retrieved August 19, 2020 from https://books.google.co.jp/books?id=1e WaveyEIlcC&pg=PA47&redir_esc=y#v=onepage&q&f=false

Lines 106-107- it is interesting that the authors scholarly acknowledged the work of Kheirallah et al 2021 to avoid release of overlapping publications with similar aims from one country within short period of time. You mentioned your work is conducted independently and you are interested to consider the prioritization of zoonotic diseases by including COVID 19 which was the limitation of Kheirallah et al. I do not think that this should not be a justifiably reason to initiate and conduct this study. If you think the work is independent, you should not raise this point, Kheirallah et al work, in the introduction as a justification for designing and conducting this study which is not actually the case. The point here is why you did not consider this work as one tool in the analysis of previously published OHZDP tools once you knew the publication before submission of this manuscript for publication. Rather, as you mentioned and discussed it should be addressed in the discussion section.

• Agree. Thank you for raising this point. As our work was conducted independently and our OHZDP tool constructed using different methods, we have removed the mention of Kheirallah et al.’s study from our introduction and instead briefly discussed their work in our discussion as you have suggested.

-Lines 123-135: The justification of considering the potential impact of increasing in number of refugees on the increasing risk of zoonotic infection is important rationale for this study which makes Jordan situation peculiar.

• Agree. We are very interested in the role of migration, conflict, and refugees, not only in regards to the risk of increasing zoonotic infections, but also what the implications are then for mitigating these risks.

-Can you explain how this study overcomes the limitation on the lack of data regarding zoonotic diseases in Jordan mentioned by Kheirallah et al. In this study, you also mentioned under study limitations lack of data on the relationship between health and peace building on the incidence of zoonoses in Jordan.

• Unfortunately, this is a limitation that we were not able to overcome; like Kheirallah et al. we too felt this to be a limitation in conducting our study. We hope that our research and other research like Kheirallah et al.’s will prompt further research and data collection within Jordan.

-Line 127. You argued that ….the groups are not inclusive enough to comprehensively represent the spectrum of issues in Jordan as the experience of refugees was not considered? If not inclusive, which sector/organization(s) should have been considered to address this gap? How your study overcome this gap?

• Thank you for this comment. This may no longer apply, as we have removed our lengthy discussion of Kheirallah et al.’s study, given our work was conducted independently. However, what we feel is missing in the OHZDP framework overall are elements of peacebuilding and development, which is specifically important in Jordan and a way to represent refugees as stakeholders. To that end, we incorporated data from the United Nations, UNHCR, and research articles which specifically focus on zoonotic diseases within refugee communities in Jordan. We were not able to overcome this gap completely, as it is difficult to obtain information that is specific to refugees, given most databases do not distinguish within their reported numbers whether the numbers are collected from Jordanian citizens or refugees. Rather, we analyzed measurement items in our confirmatory factor analysis which reflect peacebuilding and development in order to capture some of the refugee experience within construction of our tool. 

Line 147- delete the last sentence.

• Agree. We deleted this. 

Methodology

-As mentioned in the introduction, the potential rationale for this study is the under representation of the role of health and peace which can potentially influence the rank of priority zoonotic diseases in Jordan. You justified importance of the use of the proposed OHDP in countries with limited data on zoonotic diseases. Why you did not use the ODHP by considering this criteria and providing evidences, of course as a complementary to study Kheirallah et al, that show the significance of the criteria may be by allocating high weight which can potentially shift the rank of zoonotic diseases in Jordan? I do not see the significance of the thematic and statistically analysis of the previous tool used in different countries to see the similarity and predict the situation in Jordan. One of the significance of the use OHZDP is prioritizing the zoonotic diseases within the context of each country by involving the key stakeholders from various sectors and establishing inter-sectorial cooperation to combat the priority diseases.

• Unfortunately, this was not clear in our manuscript, as stated above. We did in fact use the OHZDP framework, however, we constructed our tool using different methods. We apologize for the misunderstanding and have edited to explain this more clearly. Please see lines 272-277, as well as the discussion section lines 380-416.

• We greatly appreciate the OHZDP tool and process; however, we wanted to view the issue from a different lens, especially as the situation is changing rapidly with COVID-19. Therefore, we wanted to evaluate the past OHZDP prioritization processes to understand how other countries have been prioritizing diseases in order to evaluate the necessity for changes to the tool. In addition, it was important to us to incorporate peacebuilding and development theory in constructing our OHZDP tool given the refugee situation within Jordan. Peacebuilding is a concept that is measured in many aspects. We sought to expand the One Health framework by incorporating peacebuilding and development aspects, captured by drivers of health, as seen in Table 3. 

- It would be nice if you separately present the data analysis i.e. pull all the analyses under one sub headlining-Data analysis.

• Agree. We have restricted the methodology so that all data analyses are under one sub-heading, found starting at line 165.

Line 165: To talk about sample size there should be a sampling frame from which the sample can be drawn based on selected probability or non-probability sampling technique. How, the 12 tools were selected? This is also mentioned in the limitation section. Were only 12 countries prioritized zoonotic diseases using OHDZP? The general framework in all of the 12 countries is similar in using the OHDZP with minor difference in the number of criteria and measurement scales. Can we say that they are different tools?

• We are sorry this was not clear. We edited to clarify this point. At the time of our research, the CDC website had 24 countries listed as having used the OHDZP tool to prioritize zoonoses. However, only 12 of the countries had reports publicly available. Therefore, our sample size was based on availability of reports. All 12 countries constructed their own OHZDP tools using the same framework. We think then it is fair to say the countries used the same framework and proposed workshop process, but did in fact, construct their own individual tools. Please see lines 176-180.

Line 169: We then,

• Agree. We have made this edit. 

171: The framework is the same for all where they used the WHO proposed OHZDP. Replace framework as”……groups countries based on similarity in selection of criteria and measurement questions….”

• Agree. We have made this edit. 

172. Put “to be operating under a similar framework” after Tanzania. Six countries,…

• Agree. We have made this edit. 

177: What are those three category intercepts? Specify them.

• Overall, we feel that including the lengthy description of our multiple logistic regression analysis may be confusing, as we did not use these results other than to lead us to pursuing confirmatory factor analysis. Therefore, we have focused our methodology discussion on confirmatory factor analysis and moved our logistic regression analysis description to the S2 Appendix and made the edits you have suggested there. 

o The three category intercepts were the criterion which constituted the bulk of the overall weight for most countries, which were: severity of disease, disease burden, and socioeconomic and environmental.

178-180: Why this is possible in practical sense?

• We apologize this was unclear. We have edited this in S2 Appendix to hopefully clarify our meaning. Our edit is as follows:

o We investigated if the relative criteria identified in thematic analysis (severity of disease, disease burden, socioeconomic and environmental, and control measures) weights were allocated according to the magnitude of candidate explanatory variables in addition to those three category intercepts (for example, for socioeconomic and environmental criteria we selected the following drivers of health: GDP, GINI index, and median household income): We allowed for the possibility that each country has a predetermined value (which is the value of intercept) for each category by constructing intercepts and assumed that the deviation from those predetermined values could be explained by candidate explanatory variables. 

o Essentially, none of the candidate explanatory variables moved categorical weight log odds upwards or downwards; only the intercept was found to be significant.

181-182: Re-phrase the sentence and clearly mention the dependent and independent variables for the logistic regression analysis performed. Specify the three categories and the baseline category.

• Agree. We have edited to clearly mention the dependent and independent variables, as well as the three categories (severity of disease, disease burden, and socioeconomic and environmental) and baseline category (control measures). Please S2 Appendix.

191-193: Re-phrase the sentence.

• Agree. We have rephrased this and added an explanation as to how we intentionally selected measurement items which reflect peacebuilding and development factors, by using drivers of health, such as wealth, government policies, agricultural practices, etc… We have named each measurement item and specified its connection to development / drivers of health, as indicators of peacebuilding. Please see lines 196-240.

196: How the data was scaled? Add a brief description.

• Agree. We have added a sentence describing how we scaled our data using R. Please see lines 241-242.

210-213: Only incidence data? No prevalence? Is it possible to indicate how the incidence of the diseases were diagnosed and reported (clinical, laboratory, confirmation?)

-citation for OIE database

• We have added the OIE database citation. The data collected from the OIE database only included incidence data, no prevalence, as it was documented in either number of reported cases per year. Unfortunately, it did not indicate specific criteria or information regarding how the diseases were diagnosed and reported. 

214-216: What the “immediacy” really refers in terms of measurement of diseases? Time?

-1) Delete “the disease needed to have” then, add “from animals and humans” after….span,…..

2) ….reported in humans and animals population.

• Immediacy refers to timeframe; for example, if a disease had not been seen in human populations in the last ten years, we felt that perhaps it may not warrant as much attention as diseases that have been and currently afflict humans more recently. We agree to the two changes you have suggested and have made those edits. 

217: Delete “ this resulted in in seven zoonoses” from here and take to result section.

• Agree. We have made this edit.

218-221: Including but not limited? this is confusing and please add the other search databases searched for transparency. Did you use any systematic way to search from these databases? Any key search terms used during search? Selection criteria? All these are worth mentioning.

• We apologize for the inconvenience. The other database used was the International Christian University Library database, which is comprised of a multitude of databases. We have added this for transparency and adjusted the phrasing as suggested. We did not use a systemic way to search from the databases and did not have article selection criteria, rather, we had disease selection criteria. However, we did use specific key terms while searching, and have added these terms to the article. Please see lines 158-164.

223: …this? you mean data on list of zoonoses? Mention what “ this “ refers. Here you are in another heading ( Pahse3).

• Agree. We have clarified this in our restructured methodology, which we hope will be clear now that all data analyses are under one subheading as you have suggested. 

Lines 224-225: The statement is not complete and needs re-phrasing. Indicate for what purpose the data was used. To construct modified zoonotic diseases prioritization tool? What is the central point for evaluating for the cultural compatibility between Jordan and other countries given the objective of the study (impact of violence on health)? What is the advantage over the stakeholder opinion based prioritization? Add citation for the data collected.

• Agree. We have rephrased and expanded this explanation. We have added a citation for the data collected. Please see lines 249-257.

• We incorporated data from the Hofstede Cultural Typology tool to assess cultural compatibility between Jordan and the countries, which are primarily within Africa, as cultural factors may influence decision-making behaviors. By doing this, we consider the cultural differences between Jordan and the 12 countries analyzed. This was important as we used our evaluation of past tools to assist us in constructing our own OHZDP tool. This online tool compares countries in regards to dimensions of power distance, individualism, masculinity, and uncertainty and avoidance, rating each on a 0-100 scale. We input the countries into the tool to generate a typology comparison in these dimensions. Data for Cameroon, Cote D’Ivoire, Ethiopia, Mali, and Uganda were not available from this tool. We have included results in S2 appendix. 

• The advantage of our OHZDP tool construction process versus stakeholder opinion is that we approach the problem of prioritizing zoonoses from a different perspective, one that to our knowledge, has not been attempted before. As we are not stakeholders, we have a degree of objectivity. We also collected information from a variety of sources, taking into consideration how other countries have prioritized diseases, and were able to incorporate peacebuilding and development aspects in the construction of our OHZDP tool, which we felt obligated to include given the significant refugee population. We feel the advantage is looking at the issue through a different lens, and hope that in this way, we can add to the work that experts in Jordan are already engaged in. 

Line 235: the experience of covid-19? Is it specifically mentioned in the methodology? Covid 19 should be treated as one of the diseases and evaluated based on the identified criteria and weight given.

-Start with we constructed a modified zoonotic prioritization tool

• Agree. We have made the sentence edit. We have also edited to include this in the methodology, please see lines 272-277. We did treat COVID-19 as one of the diseases, however, we felt it necessary to take into consideration the global experience of COVID-19 given it is unlike anything most of us have seen in our lifetime. We feel this experience obligates us to reconsider the way we have looked at zoonoses in the past, for example, past tools did not give much weight to socioeconomic criterion, however, given the devastating impacts of COVID-19 in this sphere, that prompted us to regard this criterion more prominently. 

Line 240-242: Again- including but not limited? this is confusing and please add the other search databases searched for transparency. Did you use any systematic way to search from these databases? Any key search terms used during search? Selection criteria? Provide website addresses for each database searched.

• We apologize for the inconvenience. The other database used was the International Christian University Library database, which is comprised of a multitude of databases. We have added this for transparency and adjusted the phrasing as suggested. Here, we did not use a systematic way to search or specific key terms. We have provided citations for our data, found in S2 and S3 appendices. 

Line 242- The criteria and the questions were identified based on the thematic and descriptive analyses. But, it seems that the authors answer (score) each questions of measurement of the criteria for each disease based on literature. I do not think that the authors can get clear information to score each questions. It would have been nice if this was validated by the stakeholders.

• We think this is a fair point as our data was limited. However, our scores were given based on the literature and data available. Given we conducted this study as objective, non-stakeholders (which differs from the usual way the OHZDP process occurs) this was the only way to collect information. We feel we were able to find sufficient information to score each of the questions; when data was not available for Jordan specifically, we sought regional data, and then global data. We have included our information sources and scoring sheets for each disease to our supplemental material for your review in S3 appendix. 

Line 246: Just before the result, mention how your question of peace/violence is addressed in the modified tool to generate a new list of priority diseases.

• Agree. We apologize this was not clear. We have started our methodology section with our theoretical framework of peacebuilding, violence, and development to explain how we incorporate this into our study. Please see lines 124-140 and 196-240.

Line 254: Indicate that the used criteria in all countries were not the same.

• Agree. We have added a sentence stating this, see line 180.

Line 255: Based on the thematic analysis, we found six criteria and 12 categorical questions (Table 1).

• Agree. We have made this edit. 

Lines 258-259: “ This suggests……….zoonotic diseases” Take to discussion section.

• Agree. We have made this edit.

Lines 259-261: Mention the countries. The fifth criteria, bioterrorism potential,….four countries (XXXX) and five countries (XXXXX).

• Agree. We have made this edit.

Lines 267-273: move to discussion section.

• Agree. We have removed this and addressed it in the newly constructed discussion section.

Line 274: Rephrase: and start as “Table 2 summarizes the criteria weighting used by each country.

• Agree. We have made this edit. 

Lines 278-280. Avoid interpretation in the result section and consider it discussion.

• Agree. We have removed this and addressed it in the newly constructed discussion section.

Line 279: Significantly (XX? quantify).

• Thank you for raising this point. This no longer applies as we have edited our results section; this line was removed as it was more suitable for the discussion section, where we have rephrased our findings. 

Line 286: Revise the table by adding row/column for the weight allotted for each criterion by each respective country and also mention in the methodology how the total weight is computed.

• Agree. We have made this edit. 

Line 290- Delete confirmatory factor analysis and rephrase as “ Table 3 shows the confirmatory factor analysis results for each one factor model fit…….”

• Agree. We have made this edit. 

Line 293: Table 3: add footnotes for SRMR, CFI& SE

• Agree. We have made this edit. 

Line 296: No heading for phase 1 results. Be consistent.

• Agree. We have ensured each phase has a heading under the results section. 

Lines 297-366: This is the result section not discussion. Present briefly in one paragraph the identified relevant zoonotic diseases and delete the detail description from here. You can provide the citations for each disease as a supplementary file. You can take some of the information from here and use them when discussing the results of phase 3 for the top priority diseases.

• Agree. We have removed these lines from the result section and instead placed an abbreviated version in the discussion. 

Lines 360-366: These should be rather part of the methodology not the result.

• Agree. We have moved this to the methodology. 

Line 367: Describe the major findings summarized in Table 4 in text before the table and refer to Table 4 for detail.

• Agree. We have summarized major findings in Table 4 before the table and added a line to refer to Table 4 for more detail. 

Lines 370-372: Already mentioned in the methodology. Here mention about the new tool focusing the findings as a result of the modified criteria and measurement/questions…..

-Be consistent for using questions Vs categorical variables Vs candidate variables.

• Agree. We have deleted the repetitive lines already stated in the methodology. In our discussion section we have provided more detail about construction of our OHZDP tool, and have included a supplementary file with detailed information (see S3 Appendix). We apologize for confusion with our terms, we have reviewed and ensured the terms for questions, categorical variables, and candidate variables are consistent. 

Lines 375-385: Take to the discussion section.

• Agree. We have moved this to the newly constructed discussion section. 

Line 389: Again the same question: Now it is clear that the criteria, the weight and the scoring of the scale for each question are based on studies in previous country according to your thematic analysis. How the authors assigned a score for each disease and prioritized the diseases in absence of stakeholder validation?

• Thank you for raising this point. As outlined above:

o Our scores were given based on the literature and data available. Given we conducted this study as objective, non-stakeholders (which differs from the usual way the OHZDP process occurs) this was the only way to collect information. We feel we were able to find sufficient information to score each of the questions; when data was not available for Jordan specifically, we sought regional data, and then global data. We have included our information sources and scoring sheets for each disease to our supplemental material for your review in S3 appendix. 

Line 391: Discussion next to Table 6

• Agree. We have added a brief description of Table 6; we have added a following discussion section as per your suggestion.

I would suggest that the authors treat the discussion section separately as a main heading by take into account the following points in sequential manner:

1. Re-iterate the purpose of the study in the first paragraph

2. 2.Followed by discussing about the new modified tool and its advantage over the Keheiriela et al study that based on the WHO proposed tool

3. 3.How the health and peace building issues are important in prioritizing zoonotic diseases under the Jordan situation

4. 4.Description about the relevant zoonotic diseases in Jordan and detail discussion only for the top priority by implicating the need for targeting these diseases to initiate prevention and control.

• Thank you for this comment and the detailed suggestions. We agree that splitting the results and discussion section would improve this paper. Therefore, we have removed the sections you pointed out in the results sections to be more suitable for the discussion, and placed them in the newly constructed discussion section. 

• Agree. We have re-iterated the purpose of the study in the first paragraph in the discussion section. We greatly appreciate your suggestion as to the sequential manner. Given our work was done independently, we have focused much less on the work by Kheirellah et al. Therefore, we decided to focus our discussion on our own findings first; we start the new discussion section with the experience of COVID-19 in Jordan, followed by details about how we constructed our OHZDP tool from our data analyses. We then compare our findings with Kheirellah et al. Finally, we tie it together with a discussion regarding peacebuilding and zoonoses in Jordan.

Lines 443-454: It is not clear how the authors try to address the impact of peace/migration on the occurrence, emergence and spread of zoonotic diseases. In the modified OHZDP, there is no data summarized in the result section that shows the importance of including drivers of health linked with violence. Can you explain this a bit further?

• We apologize for not making this link clear. We have edited to clarify our theoretical framework at the beginning of the methodology section (lines 124-140). Essentially, we define violence as social inequity (in this case, health inequity) and peacebuilding as the promotion of social justice. Development is a method of peacebuilding. Therefore, we wanted to incorporate peacebuilding by using measures of development specific to health, and so we used drivers of health to do so. The drivers of health we selected can be seen in Table 3, and we have updated our methodology to explain how each driver of health is related to peacebuilding and development, please see lines 196-140. We hope this makes it more clear and apologize it was unclear in our original manuscript. 

Line 457: The issue of health is should e situated along with the other factors such as economic crisis when it comes to the case of refuges.

• Thank you for this statement; we have edited and hope this link is clearer now. 

Line 466: Delete the discussion from here and move some of the information back to the discussion section where it fits and make your conclusion remarks separately.

• Agree. We have placed this information in the newly constructed discussion section.

Lines 472-275: I do not agree with this statement. I argue that 1) the weight allotted to each criterion cannot be considered uniform for each disease, and 2) your modified tool is also totally depended on the previously proposed tool used by different countries. May you further explain what are you communicating here?

• 1) We apologize for not making this clear. We agree that the weight allotted to each criterion cannot be considered uniform. Rather, we found weight allotment should be more evenly divided to better reflect how interconnected the impacts of zoonotic diseases can be, especially in Jordan. For example, the range among our criteria weights is 0.28 - 0.11; different from many past tools where we found ranges among criteria weighting to be as high as 0.35-0.04 (Table 2). We found distributing criteria weight more evenly across all criteria for Jordan better reflects how interconnected the impact of zoonoses can be across all spheres, as evidenced by the experiences of COVID-19. We have edited to state this in lines 410-416.

• 2) We agree. We have restated to better reflect our meaning. Throughout the paper we have now edited to try and make it clear that we did use the OHZDP framework and past tools as a foundation. We have restated this to in lines 494-497 as: To the best of our knowledge, we are the first study to connect peacebuilding and prioritizing zoonotic diseases, as we adjusted our weighting mechanism based on peacebuilding and development indicators, specifically, drivers of health relevant to Jordan and zoonotic diseases.

Reviewer 2 (Khalid A Kheirallah)

A much appreciated interdisciplinary approach to focus on OneHealth and Refugees at one time. The way COVID has but us all at risk makes it difficult for governments in developing countries to see things under the OH theme. That being said, it is critical to focus the attention on refugees first in the introduction and to provide number of refugees in Jordan. Since 2010, Jordan has accepted around 1.5 million Syrian refugees (600,000 are officially registered with UNHCR and the rest are not registered). Three main camps are in Jordan for Syrian refugees: Zaatri, Azraq are the main two within local populations, or within a short distance. Zaatri was built and made ready in days and is not organized as a camp as that for Azraq. There are real issues with clean water, sewage, and essential services. This is a real impact on OH and peacekeeping. Especially when knowing that 80% of Syrian refugees live outside camps within local (host) community. Therefore, the results may be biased towards camp residents as data on host community refugees are extremely limited. This is another major limitations for this study.

• We would like to say we are very honored that you would take the time to review our paper. We greatly appreciate the work you are doing in Jordan. We are grateful for all your comments, insights, and suggestions. 

Jordan has also accepted Iraqi refugees, and small proportions of refugees from Yemen, Libya, among others. How would this be considered into the results of the study. Also, Jordan has Palestinian refugee camps established since the 70s. Such camps have the same crowdedness and lack of clean water and sewage systems. Yet, their impact is not considered in the paper. This is another limitation for this study.

• We agree; unfortunately, much of the refugee data / literature when it comes to zoonotic and infectious diseases seems to focus on Syrian refugees. We focused on Syrian refugees specifically given they are the most recent wave of refugees and because they are the refugee population with the most data available. We did try to incorporate drivers of health and areas of health inequity in our analyses, in hopes to capture the experience of vulnerable populations. However, you are right that not specifically including the other groups of refugees is a limitation of our study. We have added this to the study limitations. 

Apparently, These limitations are much needed.

• Agree. Thank you for pointing out these limitations to our study. We have added both to the study limitation section. 

Line 415: Increase in number of TB may be due to available funding to study the disease and provide surveillance activities. How would this change the study results?

• We agree - TB likely does have more funding than many of the other neglected zoonoses on our list. Given this, the results may be skewed at this time, with bovine tuberculosis being falsely higher up on the list, simply because we do not have the data for the other diseases. The other diseases may have much higher incidences, but are unknown at this time. We hope to see more surveillance and funding for other zoonoses so that the true burden of diseases will be known.

line 466: a new zoonotic priority list could be easily established in Jordan using the same tool in Kheirallah's study. SO this statement is overestimation of the results. I think the sentence should be that this is the first tool to connect peacekeeping to zoonosis.

• Agree. We find your statement to be a more accurate assessment and have changed it accordingly. 

Line 472: Our list was based on experts within the OH field, especially MoH and MoA. They even changed the final prioritization by moving some diseases up the list based on their expertise in the field. This was not part of the presented results as it depended on theoretical experience not on-the-ground one.

• Thank you for sharing this. That makes sense; it is invaluable to have experts in the field who were able to contribute to the prioritized list and we hope our findings can add to the work already being done by experts within the country.

In table 7, under Construction of Tool, what we utilized in Kheirallah et al was a standardized CDC workshop tool that prioritized Zoonotic diseases in Jordan. This has an advantage of comparing countries' priority list and allow standardized methods for comparing results over time.

• We apologize this was unclear in our paper. We understand you used the CDC proposed workshop method. We have edited Table 7 to make this clearer to readers. 

Table 7: Prioritized List. We actually used regional prospective as Jordan is really connected to Syria, Iraq, Egypt and the other Arab states including Saudi Arabia, Qatar, and UAE. So our initial list was regional based on national needs.

• Thank you for clarifying this, we apologize for not making this clear in the paper. We have edited Table 7 to reflect this information.

Line 495: Socio-economic were a major concern during the workshop, Still, we did not see a major effect as this was conducted before COVID. If things were to be done today, SE would be a major issue and will have a higher weight.

• This is consistent with what we found in analyzing past OHZDP tools done by other countries. Often, SE was not given as much weight. We agree; prior to the experience of COVID-19 it makes sense SE would not be weighted as heavily. Part of our findings suggest that given COVID-19, SE will need to be given more weight in future constructed tools. Thank you for sharing your insight. 

Line 507: great point. Can you further elaborate and include in the major conclusion.

• Thank you for your kind words. We have elaborated per your suggestion, please see lines 428-433 and 480-487. We hope this elaboration is helpful. 

Another limitation is the utilization of data to withdraw data on zoonotic diseases in Jordan. KNowing that such data is not complete, not funded, does not have properly funded surveillance tools, makes it hard to rely on the available data to draw conclusions.

• We agree. This made it difficult for us to draw conclusions as well, given the limited availability of data. We have included this in our study limitations. We are hopeful that moving forward, there will be increased capacity to collect data on zoonotic diseases globally. 

One major comment from working with OH in Jordan is the lack of ministry of Environment in any of the OH related activities. MoH has an environmental health unit and it covered all activities related to the environment. This is a major issue that surfaced during COVID. As such, Jordan recently established the Jordan CDC. The center is expected to take the lead in OH.

• Thank you for sharing this information. It seems many infrastructures around the world have suffered issues from COVID. It is very exciting that Jordan has recently established a CDC which will take the lead in OH. We look forward to following their work. 

In the presented manuscript, the authors based their theory on numbers reported by multiple partners including Jordan MoH. During the workshop we conducted in Jordan, it was clear that surveillance data on zoonotic diseases are lacking in Jordan. For example, zoonotic diseases are treated in the camp and never make it to MoH but rather to UNHCR. The health system within the camp is not related to MoH directly except for COVID, which is a recent approach given the political covid issues.

• We came across this issue as well. We did include numbers collected from the UNHCR and the UN databases when feasible, in order to try and address this data gap and include as many refugee numbers as possible. 

In the tool that was used in Jordan, the CDC provided a guideline for which participants actually provide the 5 core questions and their scoring system. So the questions and their answers were as provided by the local stockholders. Not sure how would this change, if any, the scope of the statistical analysis used.

• Thank you for this comment. We chose our statistical analyses based on our understanding of the CDC proposed workshop, with the understanding that their proposed process involves local stakeholder participants providing the questions, scores, and answers. Unfortunately, at the time of our study, we did not have access to the local stakeholders. So in order to stay within the limit of this framework, we analyzed past studies and changed our weights accordingly based on these analyses. Essentially, we constructed our tool through different means. In addition, we wanted to evaluate the influence of peacebuilding and development measures (through analyzing specific drivers of health) and this too influenced our decision to use confirmatory factor analysis. When we came across your study at the time of submitting our paper, we were excited to see prioritization done by local experts and stakeholders and very interested to see the different results. Having local stakeholder input is invaluable. We hope that our different lens will add to the work being done.

For Hofstede Cultural Typology, what data is provided? can you provide examples of data?

• We incorporated data from the Hofstede Cultural Typology tool (found here: https://www.hofstede-insights.com/country-comparison/) to assess cultural compatibility between Jordan and the countries, which are primarily within Africa, as cultural factors may influence decision-making behaviors. By doing this, we consider the cultural differences between Jordan and the 12 countries analyzed. This was important as we used our evaluation of past tools to assist us in constructing our own OHZDP tool. This online tool compares countries in regards to dimensions of power distance, individualism, masculinity, and uncertainty and avoidance, rating each on a 0-100 scale. We input the countries into the tool to generate a typology comparison in these dimensions. Data for Cameroon, Cote D’Ivoire, Ethiopia, Mali, and Uganda were not available from this tool. We have included results in S2 appendix and we will include the graph we made from our results for your reference (graph is in the submitted attached Response to Reviewers document).

---

## [Decision Letter · Decision Letter 1]

3 Mar 2022

Situating zoonotic diseases in peacebuilding and development theories: prioritizing zoonoses in Jordan

PONE-D-21-32083R1

Dear Dr. McAlester,

We’re pleased to inform you that your manuscript has been judged scientifically suitable for publication and will be formally accepted for publication once it meets all outstanding technical requirements.

Kind regards,

Rebecca Lee Smith, D.V.M., M.S., Ph.D.

Academic Editor

PLOS ONE

Additional Editor Comments (optional):

Reviewers' comments:

Reviewer's Responses to Questions

**Comments to the Author**

1. If the authors have adequately addressed your comments raised in a previous round of review and you feel that this manuscript is now acceptable for publication, you may indicate that here to bypass the “Comments to the Author” section, enter your conflict of interest statement in the “Confidential to Editor” section, and submit your "Accept" recommendation.

Reviewer #1: (No Response)

Reviewer #2: All comments have been addressed

2. Is the manuscript technically sound, and do the data support the conclusions?

Reviewer #1: Yes

Reviewer #2: (No Response)

3. Has the statistical analysis been performed appropriately and rigorously? 

Reviewer #1: Yes

Reviewer #2: (No Response)

4. Have the authors made all data underlying the findings in their manuscript fully available?

Reviewer #1: Yes

Reviewer #2: (No Response)

5. Is the manuscript presented in an intelligible fashion and written in standard English?

Reviewer #1: No

Reviewer #2: (No Response)

6. Review Comments to the Author

Reviewer #1: The manuscript is significantly improved! I would like to acknowledge the authors for addressing the given comments and suggestions. Nonetheless, it would be so important to address the following minor comments:

1.The manuscript requires language edition. There are many instances where the statements are not correct grammatically. Just some examples,

Abstract:

-We employ (L29)

- We use (L31)

-We expand(L32)

-We undertake(L33)

-We apply(L38)

-We find(L39)

-We use (L36)

Introduction:

We write (L114)

We outline (L116) etc……. consider changing them into past tense form.

2.The data analysis only refers to the analysis of OHZDP tools. You might also pull the analysis of the literature review and the data from the Hofstede tool. Otherwise change it back to the original manuscript (Phase 2- Analysis/Evaluation of previous OHZDP tools)

3.Line 141-142; Delete “…and producing of a list of relevant zoonoses in Jordan”

4. Results and Discussion: you might consider deleting the sub-headings as you have already mentioned the phases in the methodology section. Also delete the subheading “Comparison with recent research” which is all about discussion. Lastly, I strongly recommend the publication of this manuscript after language edition.

Reviewer #2: (No Response)

7. PLOS authors have the option to publish the peer review history of their article (what does this mean?). If published, this will include your full peer review and any attached files.

Reviewer #1: **Yes: **Fanta D Gutema

Reviewer #2: **Yes: **Khalid Kheirallah

---

## [Editor Report · Acceptance letter]

9 Mar 2022

PONE-D-21-32083R1 

Situating zoonotic diseases in peacebuilding and development theories: prioritizing zoonoses in Jordan 

Dear Dr. McAlester:

I'm pleased to inform you that your manuscript has been deemed suitable for publication in PLOS ONE. Congratulations! Your manuscript is now with our production department. 

Kind regards, 

on behalf of

Dr. Rebecca Lee Smith 

Academic Editor

PLOS ONE